# Impact of self-imposed prevention measures and short-term government-imposed social distancing on mitigating and delaying a COVID-19 epidemic: A modelling study

Alexandra Teslya[1☯]*, Thi Mui Pham[1☯], Noortje G. Godijk[1☯], Mirjam E. Kretzschmar[1], Martin C. J. Bootsma[1,2], Ganna Rozhnova[1,3]

**1** Julius Center for Health Sciences and Primary Care, University Medical Center Utrecht, Utrecht University, Utrecht, The Netherlands, **2** Mathematical Institute, Utrecht University, Utrecht, The Netherlands, **3** BioISI—Biosystems & Integrative Sciences Institute, Faculdade de Ciências, Universidade de Lisboa, Lisboa, Portugal

☯ These authors contributed equally to this work.
* A.I.Teslya@umcutrecht.nl

## Abstract

### Background

The coronavirus disease (COVID-19) caused by the severe acute respiratory syndrome coronavirus 2 (SARS-CoV-2) has spread to nearly every country in the world since it first emerged in China in December 2019. Many countries have implemented social distancing as a measure to "flatten the curve" of the ongoing epidemics. Evaluation of the impact of government-imposed social distancing and of other measures to control further spread of COVID-19 is urgent, especially because of the large societal and economic impact of the former. The aim of this study was to compare the individual and combined effectiveness of self-imposed prevention measures and of short-term government-imposed social distancing in mitigating, delaying, or preventing a COVID-19 epidemic.

### Methods and findings

We developed a deterministic compartmental transmission model of SARS-CoV-2 in a population stratified by disease status (susceptible, exposed, infectious with mild or severe disease, diagnosed, and recovered) and disease awareness status (aware and unaware) due to the spread of COVID-19. Self-imposed measures were assumed to be taken by disease-aware individuals and included handwashing, mask-wearing, and social distancing. Government-imposed social distancing reduced the contact rate of individuals irrespective of their disease or awareness status. The model was parameterized using current best estimates of key epidemiological parameters from COVID-19 clinical studies. The model outcomes included the peak number of diagnoses, attack rate, and time until the peak number of diagnoses. For fast awareness spread in the population, self-imposed measures can significantly reduce the attack rate and diminish and postpone the peak number of diagnoses. We estimate that a large epidemic can be prevented if the efficacy of these measures exceeds

**Data Availability Statement:** The data used in the study are available from https://github.com/lynxgav/COVID19-mitigation.

**Funding:** This study was funded by the following: Fundação para a Ciência e a Tecnologia, project reference 131_596787873, awarded to GR, https://www.fct.pt; ZonMw 91216062, awarded to MEK, funded MEK and AT, https://www.zonmw.nl/en/; One Health European Joint Programme Horizon 2020 project 773830 (award recipient is not an author of this manuscript) funded NGG and MCJB, https://ec.europa.eu/programmes/horizon2020/en; and Aidsfonds Netherlands project P-29704 (award recipient is not an author of this manuscript) funded GR, https://aidsfonds.nl/. The funders had no role in study design, data collection and analysis, decision to publish, or preparation of the manuscript.

**Competing interests:** MEK is a member of the Editorial Board of *PLOS Medicine*. The authors have declared that no competing interests exist.

50%. For slow awareness spread, self-imposed measures reduce the peak number of diagnoses and attack rate but do not affect the timing of the peak. Early implementation of short-term government-imposed social distancing alone is estimated to delay (by at most 7 months for a 3-month intervention) but not to reduce the peak. The delay can be even longer and the height of the peak can be additionally reduced if this intervention is combined with self-imposed measures that are continued after government-imposed social distancing has been lifted. Our analyses are limited in that they do not account for stochasticity, demographics, heterogeneities in contact patterns or mixing, spatial effects, imperfect isolation of individuals with severe disease, and reinfection with COVID-19.

## Conclusions

Our results suggest that information dissemination about COVID-19, which causes individual adoption of handwashing, mask-wearing, and social distancing, can be an effective strategy to mitigate and delay the epidemic. Early initiated short-term government-imposed social distancing can buy time for healthcare systems to prepare for an increasing COVID-19 burden. We stress the importance of disease awareness in controlling the ongoing epidemic and recommend that, in addition to policies on social distancing, governments and public health institutions mobilize people to adopt self-imposed measures with proven efficacy in order to successfully tackle COVID-19.

### Author summary

#### Why was this study done?

- As of May 2020, the coronavirus disease (COVID-19) caused by the novel coronavirus (SARS-CoV-2) has spread to nearly every country in the world since it first emerged in China in December 2019.

- Confronted with a COVID-19 epidemic, public health policy makers in different countries are seeking recommendations on how to delay and/or flatten its peak.

- Evaluation of the impact of social distancing mandated by the governments in many countries and of other prevention measures to control further spread of COVID-19 is urgent, especially because of the large societal and economic impact of the former.

#### What did the researchers do and find?

- We developed a transmission model to evaluate the impact of self-imposed measures (handwashing, mask-wearing, and social distancing) due to awareness of COVID-19 and of short-term government-imposed social distancing on the epidemic dynamics.

- We showed that self-imposed measures can prevent a large epidemic if their efficacy exceeds 50%.

- We estimate that short-term government-imposed social distancing that is initiated early into the epidemic can buy time (at most 7 months for a 3-month intervention) for healthcare systems to prepare for an increasing COVID-19 burden.

- The delay to the peak number of diagnoses can be even longer and the height of the peak can be additionally reduced if the same intervention is combined with self-imposed measures that are continued after lifting government-imposed social distancing.

## What do these findings mean?

- Raising awareness of self-imposed measures such as handwashing and mask-wearing is crucial in controlling the ongoing epidemic.

- Short-term early initiated government-imposed social distancing combined with self-imposed measures provides essential time for increasing capacity of healthcare systems and can significantly mitigate the epidemic.

- In addition to policies on social distancing, governments and public health institutions should continuously mobilize people to adopt self-imposed measures with proven efficacy in order to successfully tackle COVID-19.

## Introduction

As of May 5, 2020, the novel coronavirus (SARS-CoV-2) has spread worldwide and only 13 countries have not reported any cases. It has caused over 3,640,835 confirmed cases of COVID-19 and nearly 255,100 deaths since the detection of its outbreak in China on December 31, 2019 [1]. On March 11, the World Health Organization officially declared the COVID-19 outbreak a pandemic [1]. Several approaches aimed at the containment of SARS-CoV-2 in China were unsuccessful. Airport screening of travelers was hampered by a potentially large number of asymptomatic cases and the possibility of presymptomatic transmission [2–4]. Quarantine of 14 days combined with fever surveillance was insufficient in containing the virus due to the high variability of the incubation period [5].

Now that SARS-CoV-2 has extended its range of transmission in all parts of the world, it is evident that many countries face a large COVID-19 epidemic [6]. Initial policies regarding COVID-19 prevention were mainly limited to reporting cases, strict isolation of severe symptomatic cases, home isolation of mild cases, and contact tracing [7]. However, due to the potentially high contribution of asymptomatic and presymptomatic spread [8], these case-based interventions are likely insufficient in containing a COVID-19 epidemic unless they are highly effective [8–11]. Given the rapid rise in cases and the risk of exceeding critical care bed capacities, many countries have implemented social distancing as a short-term measure aiming at reducing the contact rate in the population and, subsequently, transmission [6, 12]. Several governments have imposed nationwide partial or complete lockdowns by closing schools, public places, and nonessential businesses, canceling mass events, and issuing stay-at-home orders [6]. Previous studies on the 1918 influenza pandemic showed that such mandated interventions were effective in reducing transmission, but their timing and magnitude had a

profound influence on the course of the epidemic [13–18]. These short-term interventions were associated with a high risk of epidemic resurgence and their impact was limited if introduced too late or lifted too early [13–16].

Self-imposed prevention measures such as handwashing, mask-wearing, and social distancing could also contribute to slowing down the epidemic [19, 20]. Alcohol-based sanitizers are effective in removing the SARS coronavirus from hands [21], and handwashing with soap may have a positive effect on reducing the transmission of respiratory infections [22]. Surgical masks, often worn for their perceived protection, are not designed nor certified to protect against respiratory hazards, but they can stop droplets being spread from infectious individuals [23–25]. Information dissemination and official recommendations about COVID-19 can create awareness and motivate individuals to adopt such measures. Previous studies emphasized the importance of disease awareness for changing the course of an epidemic [26–28]. Depending on the rate and mechanism of awareness spread, the awareness process can reduce the attack rate of an epidemic or prevent it completely [26], but it can also lead to undesirable outcomes such as the appearance of multiple epidemic peaks [27, 28]. The secondary epidemic waves may appear as the result of individuals relaxing adherence to self-imposed measures prematurely in a population where the susceptible pool following the first wave is still significantly large and disease has not been completely eliminated. It is essential to assess under which conditions the spread of disease awareness that instigates self-imposed measures can be a viable strategy for COVID-19 control.

The comparison of the effectiveness of early implemented short-term government-imposed social distancing and self-imposed prevention measures on reducing the transmission of SARS-CoV-2 are currently missing but are of crucial importance in the attempt to stop its spread. If a COVID-19 epidemic cannot be prevented, it is important to know how to effectively diminish and postpone the epidemic peak to give healthcare professionals more time to prepare and react effectively to an increasing healthcare burden. Moreover, given that several countries have peaked in cases, the importance of evaluating the effect of self-imposed measures after lifting lockdown measures is profound.

Using a transmission model, we evaluated the impact of self-imposed measures (handwashing, mask-wearing, and social distancing) due to awareness of COVID-19 and of a short-term government-imposed social distancing intervention on the peak number of diagnoses, attack rate, and time until the peak number of diagnoses since the first case. We provide a comparative analysis of these interventions as well as of their combinations and assess the range of intervention efficacies for which a COVID-19 epidemic can be mitigated, delayed, or even prevented completely. Qualitatively, these results will aid public health professionals to compare and select a combination of interventions for designing effective outbreak control policies.

## Methods

### Baseline transmission model

We developed a deterministic compartmental model describing SARS-CoV-2 transmission in a population stratified by disease status (Fig 1). In this baseline model, individuals are classified as susceptible ($S$), latently infected ($E$), infectious with mild disease ($I_M$), infectious with severe disease ($I_S$), diagnosed and isolated ($I_D$), and recovered after mild or severe disease ($R_M$ and $R_S$, respectively). Susceptible individuals ($S$) can become latently infected ($E$) through contact with infectious individuals ($I_M$ and $I_S$), with the force of infection dependent on the fractions of the population in $I_M$ and $I_S$ compartments. A proportion of the latently infected individuals ($E$) will go to the $I_M$ compartment, and the remaining $E$ individuals will go to the $I_S$ compartment. We assume that infectious individuals with mild disease ($I_M$) do not require medical attention

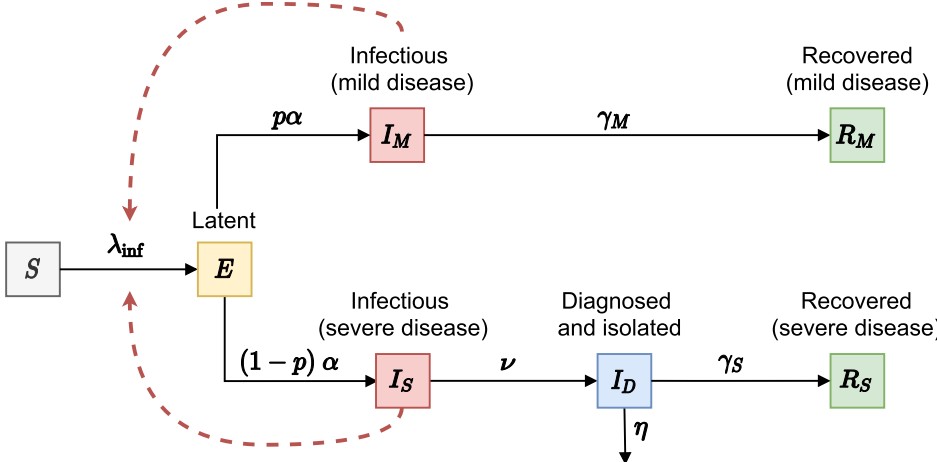

**Fig 1. Schematic of the baseline transmission model.** Black arrows show epidemiological transitions. Red dashed arrows indicate the compartments contributing to the force of infection. Susceptible persons ($S$) become latently infected ($E$) with the force of infection $\lambda_{\text{inf}}$ via contact with infectious individuals in two infectious classes ($I_M$ and $I_S$). Individuals leave the $E$ compartment at rate $\alpha$. A proportion $p$ of the latently infected individuals ($E$) will go to the $I_M$ compartment, and the proportion $(1-p)$ of $E$ individuals will go to the $I_S$ compartment. Infectious individuals with mild disease ($I_M$) recover without being conscious of having contracted COVID-19 ($R_M$) at rate $\gamma_M$. Infectious individuals with severe disease ($I_S$) are diagnosed and kept in isolation ($I_D$) at rate $\nu$ until they recover ($R_S$) at rate $\gamma_S$ or die at rate $\eta$. Table 1 provides the description and values of all parameters.

and recover ($R_M$) without being conscious of having contracted COVID-19. Infectious individuals with severe disease ($I_S$) are unable to recover without medical help, and subsequently get diagnosed and isolated ($I_D$) (in, e.g., hospitals, long-term care facilities, nursing homes) and know or suspect they have COVID-19 when they are detected. Therefore, the diagnosed compartment $I_D$ contains infectious individuals with severe disease who are both officially diagnosed and get treatment in healthcare institutions and those who are not officially diagnosed but have disease severe enough to suspect they have COVID-19 and require isolation. For simplicity, isolation of these individuals is assumed to be perfect until recovery ($R_S$), and hence they neither contribute to transmission nor to the contact process. Given the timescale of the epidemic and the lack of reliable reports on reinfections, we assume that recovered individuals ($R_M$ and $R_S$) cannot be reinfected. The infectivity of infectious individuals with mild disease is lower than the infectivity of infectious individuals with severe disease [29]. Natural birth and death processes are neglected, as the timescale of the epidemic is short compared to the mean life span of individuals. However, isolated infectious individuals with severe disease ($I_D$) may be removed from the population due to disease-associated mortality.

## Transmission model with disease awareness

In the extended model with disease awareness, the population is stratified not only by the disease status but also by the awareness status into disease-aware ($S^a$, $E^a$, $I_M^a$, $I_S^a$, $I_D^a$, and $R_M^a$) and disease-unaware ($S$, $E$, $I_M$, $I_S$, $I_D$, and $R_M$) (Fig 2A). Disease awareness is a state that can be acquired as well as lost. Disease-aware individuals are distinguished from unaware individuals in two essential ways. First, infectious individuals with severe disease who are disease-aware ($I_S^a$) get diagnosed and isolated faster ($I_D^a$), stay in isolation for a shorter period of time, and have lower disease-associated mortality than the same category of unaware individuals. The assumption we make here is that disease-aware individuals ($I_S^a$) recognize they may have COVID-19 on average faster than disease-unaware individuals ($I_S$) and get medical help

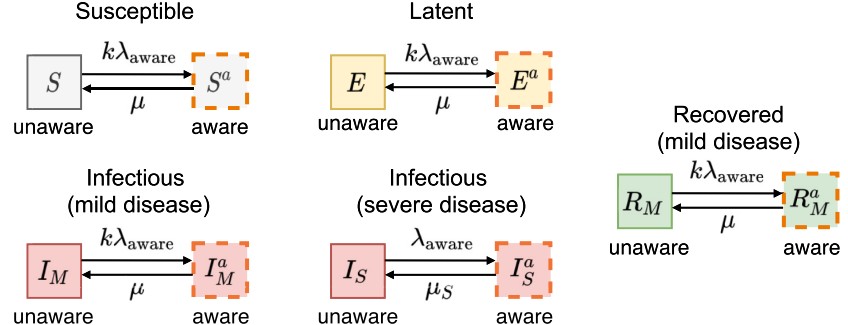

**Fig 2. Schematic of the transmission model with disease awareness.** (A) Shows epidemiological transitions in the transmission model with awareness (black arrows). The orange dashed lines indicate the compartments that participate in the awareness dynamics. The red dashed arrows indicate the compartments contributing to the force of infection. Disease-aware susceptible individuals ($S^a$) become latently infected ($E^a$) through contact with infectious individuals ($I_M$, $I_S$, $I_M^a$, and $I_S^a$) with the force of infection $\lambda_{inf}^a$. Infectious individuals with severe disease who are disease-aware ($I_S^a$) get diagnosed and isolated ($I_D^a$) at rate $\nu^a$, recover at rate $\gamma_S^a$, and die from disease at rate $\eta^a$. (B) Shows awareness dynamics. Infectious individuals with severe disease ($I_S$) acquire disease awareness ($I_S^a$) at rate $\lambda_{aware}$ proportional to the rate of awareness spread and to the current number of diagnosed individuals ($I_D$ and $I_D^a$) in the population. As awareness fades, these individuals return to the unaware state at rate $\mu_S$. The acquisition rate of awareness ($k\lambda_{aware}$) and the rate of awareness fading ($\mu$) are the same for individuals of types $S$, $E$, $I_M$, and $R_M$, where $k$ is the reduction in susceptibility to the awareness acquisition compared to $I_S$ individuals. Table 1 provides the description and values of all parameters.

earlier, which leads to a better prognosis of $I_D^a$ individuals as compared to $I_D$ individuals. Second, disease-aware individuals are assumed to use self-imposed measures such as handwashing, mask-wearing, and self-imposed social distancing that can lower their susceptibility, infectivity, and/or contact rate. Individuals who know or suspect their disease status ($I_D$, $I_D^a$, and $R_S$) do not adapt any such measures because they assume that they cannot contract the disease again. Hence, they are excluded from the awareness transition process and their behavior in the contact process is identical to disease-unaware individuals.

Similarly to Perra and colleagues [27], disease-unaware individuals acquire disease awareness at a rate proportional to the rate of awareness spread and to the current number of diagnosed individuals ($I_D$ and $I_D^a$) in the population (Fig 2B). We assume that awareness fades and individuals return to the unaware state at a constant rate. The latter means that they no longer use self-imposed measures. For simplicity, we assume that awareness acquisition and fading rates are the same for individuals of type $S$, $E$, $I_M$, and $R_M$. However, the rate of awareness acquisition is faster and the fading rate is slower for infectious individuals with severe disease ($I_S$) than for the remaining disease-aware population.

## Prevention measures

We considered short-term government intervention aimed at fostering social distancing in the population and a suite of measures that may be self-imposed by disease-aware individuals, i.e., mask-wearing, handwashing, and self-imposed social distancing.

**Mask-wearing.** Mask-wearing, while often adapted as a protective measure, may be ineffective in reducing the individual's susceptibility because laypersons, i.e., not medical professionals, are unfamiliar with correct procedures for its use (e.g., often engage in face-touching and mask adjustment) [36]. However, mask-wearing reduces infectious output [25], and therefore we assume that this measure lowers only the infectivity of disease-aware infectious individuals ($I_M^a$ and $I_S^a$) with an efficacy ranging from 0% (zero efficacy) to 100% (full efficacy).

**Handwashing.** Because infectious individuals may transmit the virus to others without direct physical contact, we assume that handwashing only reduces one's susceptibility. The efficacy of handwashing is described by the reduction in susceptibility (i.e., probability of transmission per single contact) of susceptible disease-aware individuals ($S^a$), which ranges from 0% (zero efficacy) to 100% (full efficacy). Because transmission can possibly occur through routes other than physical contact, handwashing may not provide 100% protection to those who practice it.

**Self-imposed social distancing.** Disease-aware individuals who consider themselves susceptible may also practice social distancing, i.e., maintaining distance to others and avoiding congregate settings. As a consequence, this measure leads to a change in mixing patterns in the population. The efficacy of social distancing of disease-aware individuals is described by the reduction in their contact rate, which is varied from 0% (no social distancing or zero efficacy) to 100% (complete self-isolation or full efficacy). Because contacts might not be eliminated entirely (e.g., household contacts remain), realistic values of the efficacy of self-imposed social distancing can be close to but may never reach 100%.

**Short-term government-imposed social distancing.** Governments may decide to promote social distancing policies through interventions such as school and workplace closures or by issuing stay-at-home orders and bans on large gatherings. These lockdown policies will cause a community-wide contact rate reduction, regardless of the awareness status. Here, we assume that the government-imposed social distancing is initiated if the number of diagnosed individuals exceeds a certain threshold (10–1,000 persons) and terminates after a fixed period of time (1–3 months). As such, the intervention is implemented early into the epidemic.

Government-imposed social distancing may be partial or complete depending on its efficacy, i.e., the reduction of the average contact rate in the population, which ranges from 0% (no distancing) to 100% (complete lockdown). Because during a lockdown, some contacts in the population cannot be eliminated (e.g., household contacts), realistic values of the efficacy of government-imposed social distancing can be close to but never reach 100%. For example, a 73% reduction in the average daily number of contacts was observed during the lockdown in the United Kingdom [37], but the reduction could be different in countries with more or less stringent lockdown.

## Model output

The model outputs are the peak number of diagnoses, attack rate (a proportion of the population that recovered or died after severe infection), the time to the peak number of diagnoses since the first case, and the probability of infection during the course of an epidemic (see S2 Text for a more detailed description of the latter). We compared the impact of different prevention measures and their combinations on these outputs by varying the reduction in infectivity of disease-aware infectious individuals (mask-wearing), the reduction in susceptibility of disease-aware susceptible individuals (handwashing), the reduction in contact rate of disease-aware individuals only (self-imposed social distancing), and of all individuals (government-imposed social distancing). We refer to these quantities as the efficacy of a prevention measure and vary it from 0% (zero efficacy) to 100% (full efficacy) (Table 1). The main analyses were performed for two values of the rate of awareness spread that corresponded to scenarios of slow and fast spread of awareness in the population (Table 1). For these scenarios, the proportions of the aware population at the peak of the epidemic were 40% and 90%, respectively. In the main analyses, government-imposed social distancing was initiated when 10 individuals got diagnosed and was lifted after 3 months.

Estimates of epidemiological parameters were obtained from the most recent literature (Table 1). We used contact rates for the Netherlands, but the model is appropriate for other Western countries with similar contact patterns. A detailed mathematical description of the model can be found in the S1 Text. The model was implemented in Mathematica 10.0.2.0. The code reproducing the results of this study is available at https://github.com/lynxgav/COVID19-mitigation.

## Sensitivity analyses

To allow for the uncertainty in the parameters of the baseline transmission model, we conducted sensitivity analyses with respect to the proportion of infectious individuals with mild disease, the relative infectivity of infectious individuals with mild disease, the recovery period of infectious individuals with mild disease, the delay from onset of infectiousness to diagnosis for infectious individuals with severe disease, and the basic reproduction number (see S3 Text). We also conducted sensitivity analyses for the model with disease awareness with respect to changes in the delay from the onset of infectiousness to diagnosis and isolation for disease-aware individuals, the rate of awareness spread, the relative susceptibility to awareness, and the duration of awareness (see S3 Fig). Parameter ranges used in these sensitivity analyses are specified in Table 1.

In addition, we present results for the impact on the model outcomes of all combinations of self-imposed prevention measures as their efficacy was varied from 0% to 100% and of the government-imposed social distancing, with efficacy ranging from 0% to 100%, different thresholds for initiating the intervention (1 to 1,000 diagnoses), and different durations of the intervention (3, 8, and 13 months) (see S1 Fig and S2 Fig for details).

**Table 1. Parameter values for the transmission model with and without awareness.**

| Parameters | | Value* | Source |
|---|---|---|---|
| **Epidemiological parameters** | | | |
| Basic reproduction number | $R_0$ | 2.5 (2–3) | Li and colleagues [5], Park and colleagues [30], sensitivity analyses |
| Probability of transmission per contact with $I_S$ | $\varepsilon$ | 0.048 | From $R_0 = \beta[p\sigma/\gamma_M+(1-p)/\nu]$ |
| Transmission rate of infection via contact with $I_S$ | $B$ | 0.66 per day | $\beta = c\varepsilon$ |
| Average contact rate (unique persons) | $C$ | 13.85 persons per day | Mossong and colleagues [31] |
| Relative infectivity of infectious with mild disease ($I_M$) | $\sigma$ | 50% (25%–75%) | Assumed, see, e.g., Liu and colleagues [29], sensitivity analyses |
| Proportion of infectious with mild disease ($I_M$) | $P$ | 82% (82%–90%) | Wu and colleagues [32], Anderson and colleagues [20], sensitivity analyses |
| Delay between infection and onset of infectiousness (latent period) | $1/\alpha$ | 4 days | Shorter than incubation period [5, 30, 33] |
| Delay from onset of infectiousness to diagnosis for $I_S$ | $1/\nu$ | 5 (3–7) days | Li and colleagues [5], sensitivity analyses |
| Recovery period of infectious with mild disease ($I_M$) | $1/\gamma_M$ | 7 (5–9) days | Li Xingwang[†], sensitivity analyses |
| Delay from diagnosis to recovery for unaware diagnosed ($I_D$) | $1/\gamma_S$ | 14 days | WHO [34] |
| Relative infectivity of isolated ($I_D$) | | 0% | Assuming perfect isolation |
| Case fatality rate of unaware diagnosed ($I_D$) | $f$ | 1.6% | Althaus and colleagues [35] Park and colleagues [30] |
| Disease-associated death rate of unaware diagnosed ($I_D$) | $\eta$ | 0.0011 per day | $\eta = \gamma_S f/(1-f)$ |
| **Awareness parameters** | | | |
| Rate of awareness spread (slow, fast and range) | $\delta$ | $5\times10^{-5}$, 1 ($10^{-6}$ − 1) per year | Assumed, sensitivity analyses |
| Relative susceptibility to awareness acquisition for $S$, $E$, $I_M$, and $R_M$ | $k$ | 50% (0%–100%) | Assumed, sensitivity analyses |
| Duration of awareness for $S^a$, $E^a$, $I_M^a$, and $R_M^a$ | $1/\mu$ | 30 (7–365) days | Assumed, sensitivity analyses |
| Duration of awareness for $I_S^a$ | $1/\mu_S$ | 60 (7–365) days | Longer than $1/\mu$, sensitivity analyses |
| Delay from onset of infectiousness to diagnosis for $I_S^a$ | $1/\nu^a$ | 3 (1–5) days | Shorter than $1/\nu$, sensitivity analyses |
| Delay from diagnosis to recovery of aware diagnosed ($I_D^a$) | $1/\gamma_S^a$ | 12 days | Shorter than $1/\gamma_S$ |
| Case fatality rate of aware diagnosed ($I_D^a$) | $f^a$ | 1% | Smaller than $f$ |
| Disease-associated death rate of aware diagnosed ($I_D^a$) | $\eta^a$ | 0.0008 per day | $\eta = \gamma_S^a f^a/(1 - f^a)$ |
| **Prevention measure parameters** | | | |
| Efficacy of mask-wearing (reduction in infectivity) | | 0%–100% | Varied |
| Efficacy of handwashing (reduction in susceptibility) | | 0%–100% | Varied |
| Efficacy of self-imposed contact rate reduction | | 0%–100% | Varied |
| Efficacy of government-imposed contact rate reduction | | 0%–100% | Varied |
| Duration of government-imposed social distancing | | 3 (1–13) months | Assumed, sensitivity analyses |

(*Continued*)

**Table 1.** (Continued)

| Parameters | Value* | Source |
|---|---|---|
| Threshold for initiation of government-imposed social distancing | 10 (1–1,000) diagnoses | Assumed, sensitivity analyses |

*Mean or median values were used from literature; range was used in the sensitivity analyses.

†Expert at China's National Health Commission.

## Results

Our analyses show that disease awareness spread has a significant effect on the model predictions. We first considered the epidemic dynamics in a disease-aware population where handwashing is promoted, as an example of self-imposed measures (Fig 1). Then, we performed a systematic comparison of the impact of different prevention measures on the model output for slow (Fig 2) and fast (Fig 3) rate of awareness spread.

### Epidemic dynamics

All self-imposed measures and government-imposed social distancing have an effect on the COVID-19 epidemic dynamics. The qualitative and quantitative impact, however, depends strongly on the prevention measure and the rate of awareness spread. The baseline model predicts 46 diagnoses per 1,000 individuals at the peak of the epidemic, an attack rate of about 16%, and the time to the peak of about 5.2 months (red line, Fig 3A and 3B). In the absence of prevention measures, a fast spread of disease awareness reduces the peak number of diagnoses by 20% but has only a minor effect on the attack rate and peak timing (orange line, Fig 3A and 3B). This is expected, as disease-aware individuals with severe disease seek medical care sooner and therefore get diagnosed faster, causing fewer new infections as compared to the baseline model. Awareness dynamics coupled with the use of self-imposed prevention measures has an even larger impact on the epidemic. The blue line in Fig 3A shows the epidemic curve for the scenario when disease-aware individuals use handwashing as a self-imposed prevention measure. Even if the efficacy of handwashing is modest (i.e., 30% as in Fig 3A), the impact on the epidemic can be significant; namely, we predicted a 65% reduction in the peak number of diagnoses, a 29% decrease in the attack rate, and a delay in peak timing of 2.7 months (Fig 3A and 3B).

The effect of awareness on the disease dynamics can also be observed in the probability of infection during the course of the epidemic. In the model with awareness and no measures, the probability of infection is reduced by 4% for all individuals. Handwashing with an efficacy of 30% reduces the respective probability by 14% for unaware individuals and by 29% for aware individuals. Note that the probability of infection is highly dependent on the type of prevention measure. The detailed analysis is given in S2 Text.

### A comparison of prevention measures

**Slow spread of awareness.** Fig 4 shows the impact of all considered self-imposed measures as well as of the government-imposed social distancing on the peak number of diagnoses, attack rate, and the time to the peak for a slow rate of awareness spread. In this scenario, the model predicts progressively larger reductions in the peak number of diagnoses and in the attack rate as the efficacy of the self-imposed measures increases. In the limit of 100% efficacy, the reduction in the peak number of diagnoses is 23% to 30% (Fig 4A) and the attack rate decreases from 16% to 12%–13% (Fig 4B). The efficacy of the self-imposed measures has very little impact on the peak timing when compared to the baseline; i.e., no awareness in the

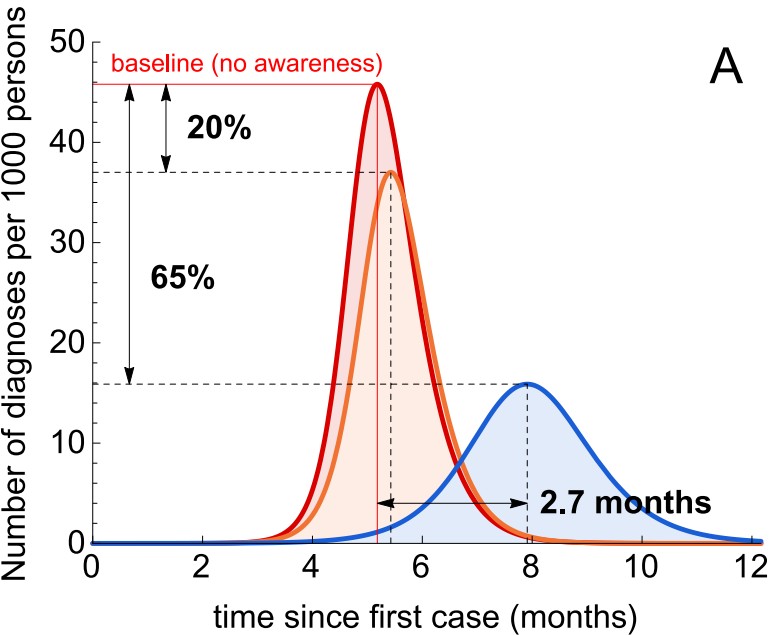

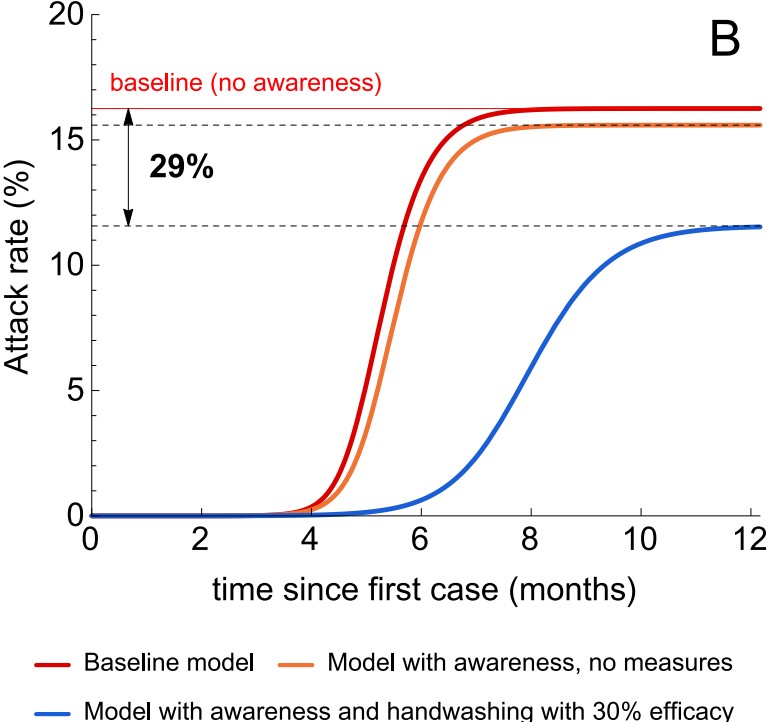

**Fig 3. Illustrative simulations of the transmission model.** (A, B) Shows the number of diagnoses and the attack rate during the first 12 months after the first case under three model scenarios. The red lines correspond to the baseline transmission model. The orange lines correspond to the model with a fast rate of awareness spread and no interventions. The blue lines correspond to the latter model, where disease awareness induces the uptake of handwashing with an efficacy of 30%.

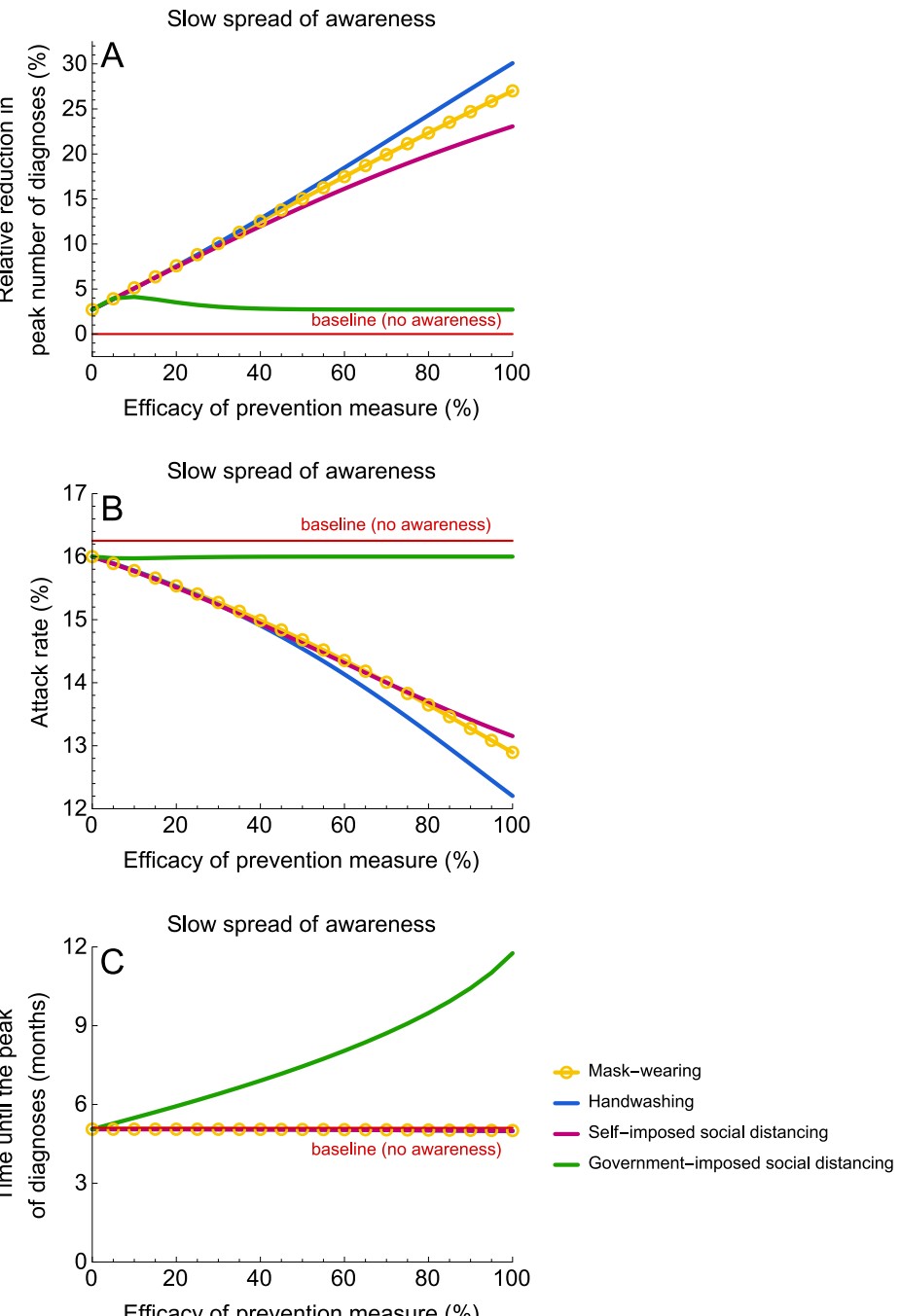

**Fig 4. Impact of prevention measures on the epidemic for a slow rate of awareness spread.** (A–C) Shows the relative reduction in the peak number of diagnoses, the attack rate (proportion of the population that recovered or died after severe infection), and the time until the peak number of diagnoses. The efficacy of prevention measures was varied between 0% and 100%. In the context of this study, the efficacy of social distancing denotes the reduction in the contact rate. The efficacy of handwashing and mask-wearing are given by the reduction in susceptibility and infectivity, respectively. The simulations were started with one case. Government-imposed social distancing was initiated after 10 diagnoses and lifted after 3 months. For parameter values, see Table 1. Please note that the blue line corresponding to handwashing is not visible in (C) because it almost completely overlaps with lines for mask-wearing and self-imposed social distancing.

population (Fig 4C). Because the proportion of aware individuals who change their behavior is too small to make a significant impact on transmission, self-imposed measures can only mitigate but not prevent an epidemic.

When awareness spreads at a slow rate, a 3-month government intervention has a contrasting impact to the self-imposed measure scenario. The time to the peak number of diagnoses is longer for more stringent contact rate reductions. For example, a complete lockdown (government-imposed social distancing with 100% efficacy) can postpone the peak by almost 7 months, but its magnitude and attack rate are unaffected (with respect to the baseline model without measures and awareness). Similar predictions are expected, as long as government-imposed social distancing starts early (e.g., after tens to hundreds of cases) and is lifted a few weeks to a few months later. This type of intervention halts the epidemic for the duration of intervention, but, because of a large pool of susceptible individuals, epidemic resurgence is expected as soon as social distancing measures are lifted.

**Fast spread of awareness.**   Because the government intervention reduces the contact rate of all individuals irrespective of their awareness status, it has a comparable impact on transmission for scenarios with fast and slow rate of awareness spread (compare Fig 4 and Fig 5). However, the impact of self-imposed measures is drastically different when awareness spreads fast. All self-imposed measures are more effective than the short-term government intervention. These measures not only reduce the attack rate (Fig 5B) and diminish and postpone the peak number of diagnoses (Fig 5A and 5C), but they can also prevent a large epidemic altogether when their efficacy is sufficiently high (about 50%). Note that when the rate of awareness is fast, as the number of diagnoses grows, the population becomes almost homogeneous, with most individuals being disease-aware. It can be shown that in such populations, prevention measures yield comparable results if they have the same efficacy.

**Combinations of prevention measures.**   If government-imposed social distancing is combined with a self-imposed prevention measure, the model predicts that the relative reductions in the peak number of diagnoses and attack rate are determined by the efficacy of the self-imposed measure, whereas the timing of the peak is determined by the efficacies of both the self-imposed measure and the government intervention. This is demonstrated in Fig 6, where we used a combination of handwashing with efficacies of 30%, 45%, and 60%, and government-imposed social distancing with efficacy ranging from 0% to 100% for slow and fast spread of awareness. Our results show that the effect of the combined intervention highly depends on the rate of awareness spread. Fast awareness spread is crucial for a large reduction in the peak number of diagnoses (Fig 6A) and in the attack rate (Fig 6B). Note, that for fast spread of awareness, a combination of a complete lockdown and handwashing with an efficacy of 30% could postpone the time to the peak number of diagnoses by nearly 10 months (Fig 6C). Thus, when combined with short-term government-imposed social distancing, handwashing can contribute to mitigating and delaying the epidemic after the lockdown is relaxed. The second wave of the epidemic could be prevented completely if the efficacy of handwashing exceeds 50% (Fig 6A). The results for the combination of mask-wearing and government-imposed social distancing are similar.

The effect of combinations of self-imposed measures (e.g., handwashing and mask-wearing) is additive (see S1 Fig). This means that, for fast spread of awareness, a large outbreak can be prevented by, for example, a combination of handwashing and self-imposed social distancing, each with an efficacy of around 25% (or other efficacies adding up to 50%).

## Discussion

For many countries around the world, the focus of public health officers in the context of COVID-19 epidemic has shifted from containment to mitigation and delay. Our study

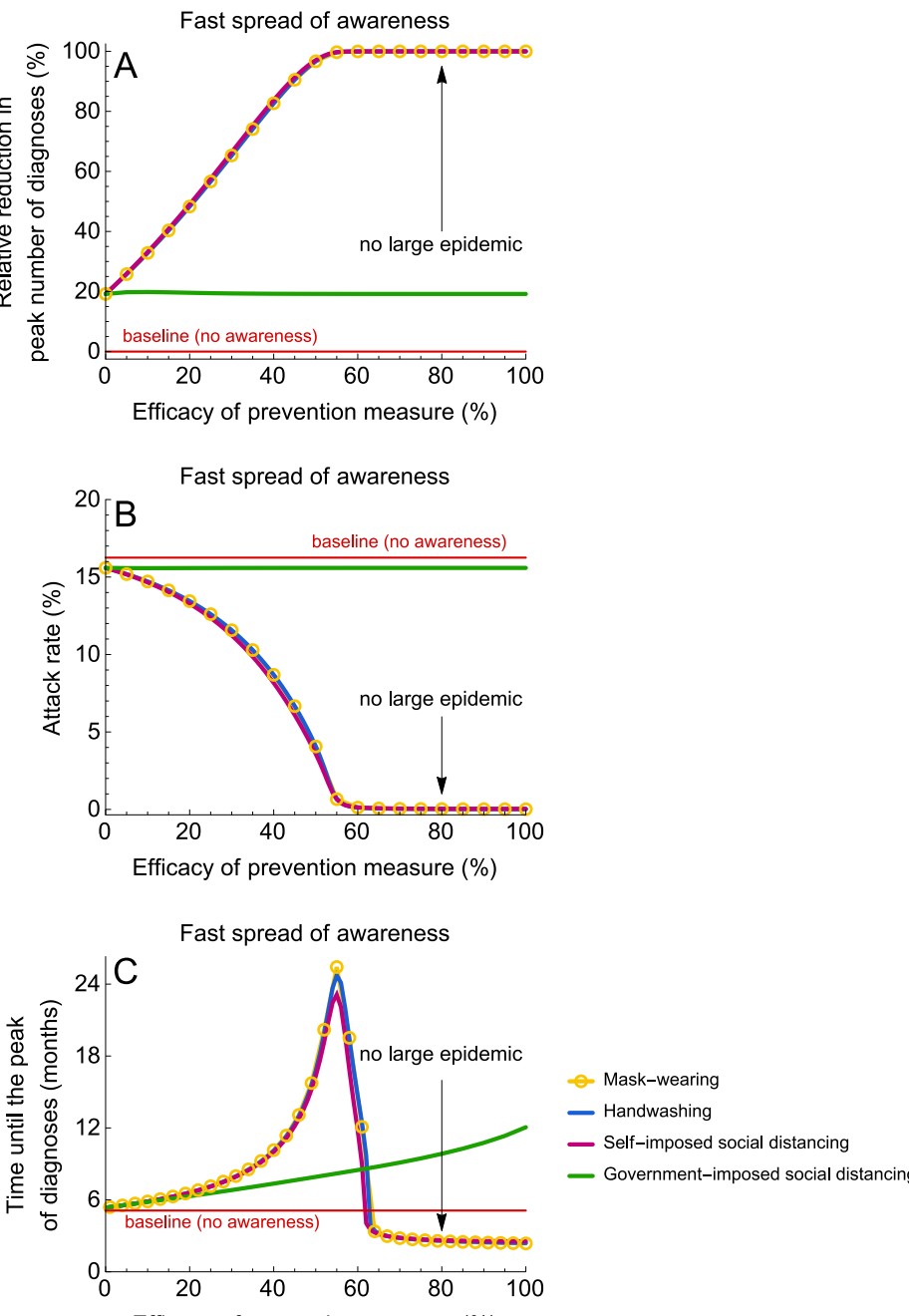

**Fig 5. Impact of prevention measures on the epidemic for a fast rate of awareness spread.** (A–C) Shows the relative reduction in the peak number of diagnoses, the attack rate (proportion of the population that recovered or died after severe infection), and the time until the peak number of diagnoses. The efficacy of prevention measures was varied between 0% and 100%. In the context of this study, the efficacy of social distancing denotes the reduction in the contact rate. The efficacy of handwashing and mask-wearing are given by the reduction in susceptibility and infectivity, respectively. The simulations were started with one case. Government-imposed social distancing was initiated after 10 diagnoses and lifted after 3 months. For parameter values, see Table 1. Please note that the blue line corresponding to handwashing is not visible in (A) because it almost completely overlaps with lines for mask-wearing and self-imposed social distancing.

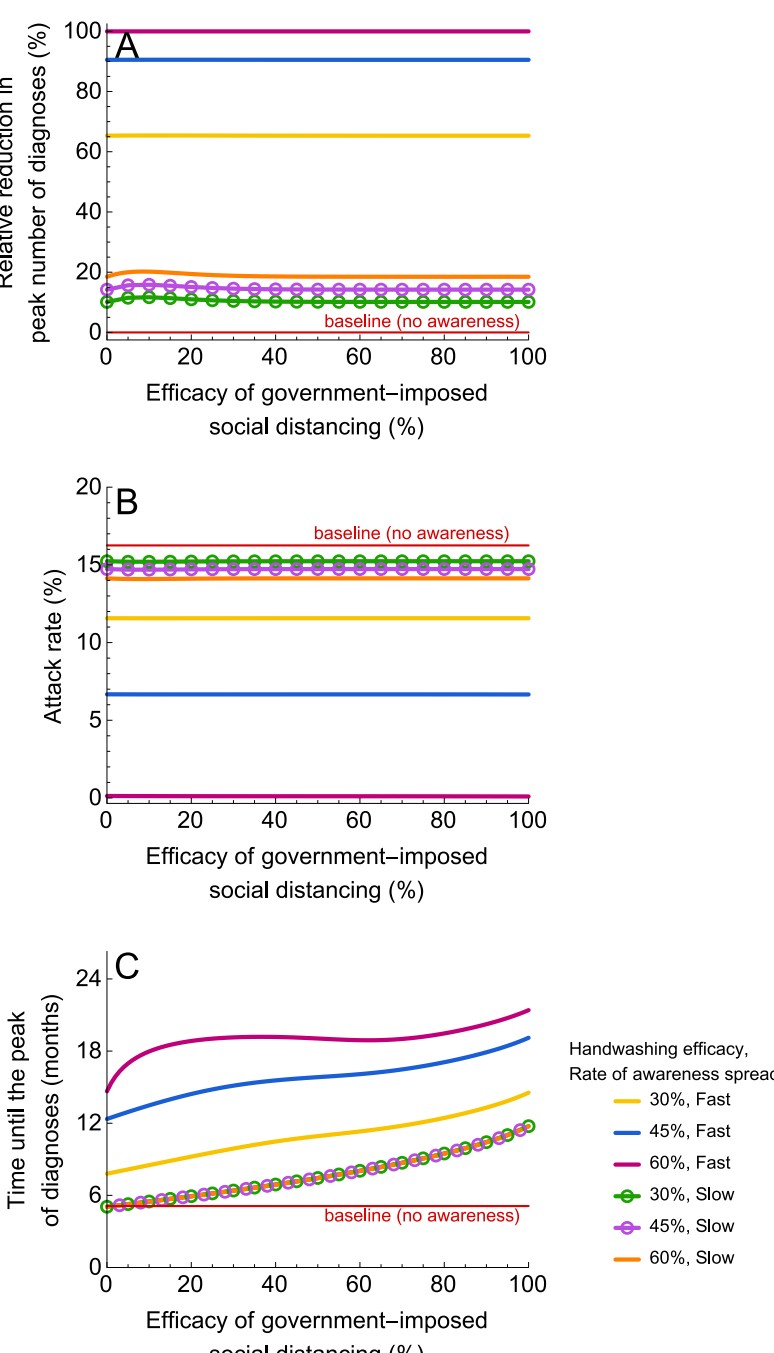

**Fig 6. Impact on the epidemic of a combination of government-imposed social distancing and handwashing.** (A–C) Shows the relative reduction in the peak number of diagnoses, the attack rate (proportion of the population that recovered or died after severe infection), and the time until the peak number of diagnoses. The efficacy of handwashing was 30%, 45%, and 60%. In the context of this study, the efficacy of social distancing denotes the reduction in the contact rate. The efficacy of handwashing is given by the reduction in susceptibility. The simulations were started with one case. Government-imposed social distancing was initiated after 10 diagnoses and lifted after 3 months. For parameter values, see Table 1.

provides new insights for designing effective outbreak control strategies. Based on our results, we conclude that handwashing, mask-wearing, and social distancing adopted by disease-aware individuals can delay the epidemic peak, flatten the epidemic curve, and reduce the attack rate. We show that the rate at which disease awareness spreads has a strong impact on how self-imposed measures affect the epidemic. For a slow rate of awareness spread, self-imposed measures have less impact on transmission, as not many individuals adopt them. However, for a fast rate of awareness spread, their impact on the magnitude and timing of the peak increases with increasing efficacy of the respective measure. For all measures, a large epidemic can be prevented when the efficacy exceeds 50%. Moreover, the effect of combinations of self-imposed measures is additive. In practical terms, it means that SARS-CoV-2 will not cause a large outbreak in a country where 90% of the population adopts handwashing and social distancing that are 25% efficacious (i.e., reduce susceptibility and contact rate by 25%, respectively).

Although our analyses indicate that the effects of self-imposed measures on mitigating and delaying the epidemic for the same efficacies are similar (see Fig 4 and Fig 5), not all explored efficacy values may be achieved for each measure. Wong and colleagues [22] and Cowling and colleagues [24] performed a systematic review and meta-analysis on the effect of handwashing and face masks on the risk of influenza virus infections in the community. While the authors highlight the potential importance of both hand hygiene and face masks, only modest effects could be ascertained with a pooled risk ratio of 0.73 (95% CI 0.6–0.89) for a combination of these two measures. However, the authors also highlight the small number of randomized controlled trials and the heterogeneity of the studies as notable limitations that may have led to these results. Given the high uncertainty around the efficacies of hand hygiene and mask-wearing on their own, the promotion of a combination of these measures might become preferable to recommending handwashing or mask-only measures. For self-imposed social distancing, contacts might not be eliminated entirely (e.g., household contacts remain), and therefore realistic values of the efficacy of self-imposed social distancing can be close to but may never reach 100%. Thus, for a fair comparison between measures, realistic efficacy values of a specific measure should be taken into consideration.

We contrasted self-imposed measures stimulated by disease awareness with mandated social distancing. Our analyses show that short-term government-imposed social distancing that is implemented early into the epidemic can delay the epidemic peak but does not affect its magnitude nor the attack rate. For example, a complete lockdown of 3 months imposing a community-wide contact rate reduction that starts after tens to thousands diagnoses in the country can postpone the peak by about 7 months. Such an intervention is highly desirable, when a vaccine is being developed or when healthcare systems require more time to treat cases or increase capacity. If this intervention is implemented in a population that exercises a self-imposed measure that continues to be practiced even after the lockdown is over, then the delay can be even longer (e.g., up to 10 months for handwashing with 30% efficacy). In the context of countries that implemented social distancing as a measure to "flatten the curve" of the ongoing epidemics, peaked in cases, and are now planning or have already started gradual lifting of social distancing, it means that governments and public health institutions should intensify the promotion of self-imposed measures to diminish and postpone the peak of the potential second epidemic wave. The potential second wave could be prevented altogether if the coverage of a self-imposed measure in the population and its efficacy are sufficiently high (e.g., 90% and 50%, respectively). Our sensitivity analyses showed that lower or higher efficacies can be required to prevent a large epidemic for countries with smaller or larger basic reproduction numbers (see S3 Text).

Since for many countries the COVID-19 epidemic is still in its early stages, government-imposed social distancing was modelled as a short-term intervention initiated when the

number of diagnosed individuals was relatively low. Our sensitivity analyses showed that government interventions introduced later into the epidemic (at 100–1,000 diagnoses) and imposed for a longer period of time (3–13 months) not only delay the peak of the epidemic but also reduce it for intermediate efficacy values (see S2 Fig). Previous studies suggested that the timing of mandated social distancing is crucial for its viability in controlling a large disease outbreak [13, 14, 16, 38]. As discussed by Hollingsworth and colleagues [16] and Anderson and colleagues [20], a late introduction of such interventions may have a significant impact on the epidemic peak and attack rate. However, the authors also showed that the optimal strategy is highly dependent on the desired outcome. A detailed analysis of government intervention with different timings and durations that also takes into account the economic and societal consequences, and the cost of SARS-CoV-2 transmission is a subject for future work.

To our knowledge, our study is the first to provide comparative analysis of a suite of self-imposed measures, government-imposed social distancing, and their combinations as strategies for mitigating and delaying a COVID-19 epidemic. Several studies (e.g., [39–42]) looked at the effect of different forms of social distancing, but they did not include self-imposed measures such as handwashing and mask-wearing. Some of these studies concluded that one-time social-distancing interventions will be insufficient to maintain COVID-19 prevalence within the critical care capacity [40, 42]. In our analyses, we explored the full efficacy range for all self-imposed prevention measures and different durations and thresholds for initiation of government intervention. Our results allow drawing conclusions on which combination of prevention measures can be most effective in diminishing and postponing the epidemic peak when realistic values for the measure's efficacy are taken into account. We showed that spreading disease awareness such that highly efficacious preventive measures are quickly adopted by individuals can be crucial in reducing SARS-CoV-2 transmission and preventing a large epidemics of COVID-19.

Our model has several limitations. It does not account for stochasticity, demographics, heterogeneities in contact patterns, spatial effects, inhomogeneous mixing, imperfect isolation of individuals with severe disease, and reinfection with COVID-19. Our conclusions can therefore be drawn on a qualitative level. Detailed models will have to be developed to design and tailor effective strategies in particular settings. The impact of the duration of immunity has been explored by Kissler and colleagues [43]. The effect of nonpermanent immunity on the results of our model would be an interesting subject for future work. To take into account the uncertainty in SARS-CoV-19 epidemiological parameters, we performed sensitivity analyses to test the robustness of the model predictions. As more data become available, our model can be easily updated. In addition, our study assumes that individuals become disease-aware with a rate of awareness acquisition proportional to the number of currently diagnosed individuals. Other forms for the awareness acquisition rate that incorporate, e.g., the saturation of awareness, may be more realistic and would be interesting to explore in future studies. Furthermore, we assume that handwashing may reduce the susceptibility of an individual down to 0% and therefore neglect aerosol transmission of SARS-CoV-2. Thus, the impact of handwashing on the epidemic may be an overestimation. However, while there is preliminary evidence on SARS-CoV-2 RNA detection in aerosols [44], there is still uncertainty about the level of infectiousness of the detected aerosols and the significance of potential airborne transmission. Current recommendations by the World Health Organization are still focused on droplet and contact precautions [45]. Our model may be adapted as more information on the relative contribution of the transmission routes of COVID-19 emerges.

In conclusion, we provide the first empirical basis of how stimulating the uptake of effective prevention measures, such as handwashing or mask-wearing, combined with government-imposed social distancing intervention, can be pivotal to achieving control over a COVID-19

epidemic. While information on the rising number of COVID-19 diagnoses reported by the media may fuel anxiety in the population, wide and intensive promotion of self-imposed measures with proven efficacy by governments or public health institutions may be a key ingredient to tackle COVID-19.

## Supporting information

**S1 Text. Mathematical description of the model.**
(PDF)

**S2 Text. Impact of awareness process on the probability of infection.**
(PDF)

**S3 Text. Sensitivity analyses of the baseline transmission model.**
(PDF)

**S1 Fig. Impact of combinations of self-imposed prevention measures.** Top and bottom panels show the peak number of diagnoses and time until the peak of diagnoses (months) since the first case for all combinations of self-imposed prevention measures as their efficacy was varied from 0% to 100%. The figures were obtained for a fast rate of awareness spread and population of 17 million individuals. The model predicts that the effect of combinations of self-imposed measures is additive. This means that a large outbreak can be prevented by, for example, a combination of handwashing and self-imposed social distancing, each with an efficacy of around 25% (or other efficacies adding up to 50%).
(PDF)

**S2 Fig. Impact of government-imposed social distancing interventions.** Top and bottom figures show the peak number of diagnoses and time until the peak of diagnoses (months) since the first case for interventions that last 3, 8, and 13 months (left, middle, and right row, respectively). The efficacy of government-imposed contact rate reduction was varied from 0% to 100% (y-axis), and the threshold for initiation of intervention was between 1 and 1,000 diagnoses (x-axis). The figures were obtained for a fast rate of awareness spread and population of 17 million individuals. The model predicts that government intervention introduced later into the epidemic and imposed for a longer period of time not only delays the peak of the epidemic but also reduces it for intermediate efficacy values.
(PDF)

**S3 Fig. Sensitivity analyses of the transmission model with disease awareness.** Page 1: The analyses demonstrate that the transmission model with disease awareness is sensitive with respect to changes in the delay from the onset of infectiousness to diagnosis for disease-aware individuals and to the rate of awareness spread. Page 2: The analyses demonstrate that the transmission model with disease awareness is not sensitive with respect to changes in the relative susceptibility to awareness and duration of awareness.
(PDF)

## Acknowledgments

This study has benefited greatly from the feedback provided by the Infectious Disease Modelling group based at the Julius Center for Health Sciences and Primary Care, UMC Utrecht, Utrecht University.

## Author Contributions

**Conceptualization:** Alexandra Teslya, Thi Mui Pham, Noortje G. Godijk, Mirjam E. Kretzschmar, Martin C. J. Bootsma, Ganna Rozhnova.

**Formal analysis:** Alexandra Teslya, Ganna Rozhnova.

**Funding acquisition:** Mirjam E. Kretzschmar.

**Investigation:** Alexandra Teslya, Thi Mui Pham, Ganna Rozhnova.

**Methodology:** Alexandra Teslya, Thi Mui Pham, Noortje G. Godijk, Ganna Rozhnova.

**Project administration:** Thi Mui Pham.

**Software:** Alexandra Teslya, Ganna Rozhnova.

**Supervision:** Ganna Rozhnova.

**Visualization:** Alexandra Teslya, Ganna Rozhnova.

**Writing – original draft:** Alexandra Teslya, Thi Mui Pham, Noortje G. Godijk, Ganna Rozhnova.

**Writing – review & editing:** Alexandra Teslya, Thi Mui Pham, Noortje G. Godijk, Mirjam E. Kretzschmar, Martin C. J. Bootsma, Ganna Rozhnova.

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
