## [Editor Report · Decision Letter 0]

24 Mar 2020

Dear Dr Teslya, 

Thank you for submitting your manuscript entitled "Impact of self-imposed prevention measures and short-term government intervention on mitigating and delaying a COVID-19 epidemic" for consideration by PLOS Medicine.

Your manuscript has now been evaluated by the PLOS Medicine editorial staff [as well as by an academic editor with relevant expertise] and I am writing to let you know that we would like to send your submission out for external peer review.

Kind regards,

Adya Misra, PhD,

Senior Editor

PLOS Medicine

---

## [Decision Letter · Decision Letter 1]

23 Apr 2020

Dear Dr. Teslya,

Thank you very much for submitting your manuscript "Impact of self-imposed prevention measures and short-term government intervention on mitigating and delaying a COVID-19 epidemic" (PMEDICINE-D-20-00973R1) for consideration at PLOS Medicine. 

[LINK]

In light of these reviews, I am afraid that we will not be able to accept the manuscript for publication in the journal in its current form, but we would like to consider a revised version that addresses the reviewers' and editors' comments. Obviously we cannot make any decision about publication until we have seen the revised manuscript and your response, and we plan to seek re-review by one or more of the reviewers. 

We expect to receive your revised manuscript by May 07 2020 11:59PM. Please email us (plosmedicine@plos.org) if you have any questions or concerns.

We look forward to receiving your revised manuscript. 

Sincerely,

Adya Misra, PhD

Senior Editor 

PLOS Medicine

plosmedicine.org

Title-Please revise your title according to PLOS Medicine's style. Your title must be nondeclarative and not a question. It should begin with main concept if possible. "Effect of" should be used only if causality can be inferred, i.e., for an RCT. Please place the study design ("A randomized controlled trial," "A retrospective study," "A modelling study," etc.) in the subtitle (ie, after a colon).

Abstract

Background- please explicitly state the aim of your study here. You may wish to update this section to reflect the current situation 

Methods and findings- please provide brief info about model parameters and assumptions 

Methods and findings- the last sentence of this section should be a limitation of your study design

Conclusions-please begin the section with “our results suggest” or similar

Author summary

At this stage, we ask that you add bullet points to the Author Summary. The Author Summary should immediately follow the Abstract in your revised manuscript. This text is subject to editorial change. Please see our author guidelines for more information: https://journals.plos.org/plosmedicine/s/revising-your-manuscript#loc-author-summary. 

Introduction

Please update dates and numbers as you see fit

Lines 69-75, you may wish to update this as several countries may have peaked in cases and perhaps focus on how these measures may affect lockdown easing which is topical right now

Methods

Could you clarify the extent of government mandated social distancing accounted for in your models? For example if this is a partial intervention or complete lockdown 

Please can you provide further details of the sensitivity analysis in the methods section

While none of the reviewers have specifically asked for the models to be validated using current data, please let us know if there are reasons to not do this during revision. 

Discussion

Line 274- please add “to our knowledge” or similar

Please update your discussion incorporating recent relevant findings as you see fit

Bibliography

Please use Vancouver style

Please ensure that the study is reported according to the STROBE guideline, and include the completed [STROBE or other] checklist as Supporting Information. When completing the checklist, please use section and paragraph numbers, rather than page numbers. Please add the following statement, or similar, to the Methods: "This study is reported as per the Strengthening the Reporting of Observational Studies in Epidemiology (STROBE) guideline (S1 Checklist)."

Please report your study according to the relevant guideline, which can be found here: http://www.equator-network.org/

Comments from the reviewers:

Reviewer #1: This article presents a mathematical model of the current novel coronavirus outbreak and focuses on the impact that handwashing, mask-wearing, and social distancing can have on transmission. While the results are informative, the lack of uncertainty around them and around the parameter inputs, makes it difficult to see how likely the results allow us to make accurate predictions about what will happen if we implement these measures. That is my main concern. My other main concern is that this models various strategies that are already being used in complement. Regardless of what intervention we try, some people will implement better handwashing and some people will self-isolate and this paper treats the all independent with one exception (hand washing at 30% efficacy and government imposed distancing).

Other comments about the manuscript are below. 

Just to be as current as you can, I would suggest the authors update the opening sentence to the latest date possible before this is published.

In line 50, I think the phrase "expected developments in the next few weeks," is already out of date, and now these things have happened, so I would update.

In line 52 you say "Governments can impose social distancing by closing schools or public places, cancelling mass events, and promoting remote work" they are now issuing stay at home orders, in the US "shelter in place" orders.

The sentence in line 53 is vague and I think it is important "Previous studies showed that the timing and magnitude of such mandated interventions had a profound influence on the 1918 influenza pandemic." Specifically acting early made a big difference"

The assumption in the SEIR model is that all sick individual with severe symptoms are diagnosed, but this isn't likely to be the case. More likely only a percentage of them are even if that percentage is high.

The model also assumes that all individual who are in isolation stay there until they are no longer infectious, which is also unlikely to be true, some will continue to have contact, particularly within households.

I do support the decision to model no reinfection but it would be important to note that it is not yet known whether this is true 100% of the time.

It isn't clear to me how this model deal with the fact that those who develop severe symptoms often develop mild symptoms first and an transmit the virus during that time and also can transmit before they have any symptoms. The authors do say "However, severely symptomatic patients in isolation may be removed from the population due to disease-associated mortality." So I assume it may be accounted for but I'm not clear on how.

It is unclear t me why disease aware individuals stay insolation for a shorter period of time than those who are not aware.

The table of parameters is very nice. I appreciate that the authors varied the parameters in the effectiveness, but the other parameters in the model are assumptions and we don't know then to be true. Adding in the uncertainty in those parameters would allow for uncertainty in the results. 

Varying efficacy from 0 to 100% for the interventions seems like an overly wide range. We don't think they would ever be 0 or 100% effective so what would be plausible ranges? And what evidence backs those estimates up?

Reviewer #2: Teslya et al write a really interesting piece on the impact of self-imposed and government prevention measures on controlling COVID-19 epidemic. I found the manuscript very well written and the mathematical models carefully described. 

I only have minor comments:

Abstract:

Lines 15-17 "Government-imposed social distancing introduced later into the epidemic and kept for a longer period of time not only delays the peak number of diagnoses but also reduces it for intermediate efficacy values" This is not one of the main findings of the article (and it has already been shown in previous articles). This sentence is mentioned only in the Abstract and shown in the Supplementary material but in the main analysis of the paper both timing and duration of government-imposed social distancing are fixed. Therefore, I would suggest removing this sentence from the Abstract.

Author summary: 

Line 26 All figures need to be updated

Introduction: 

First paragraph (lines 37-43) needs to be updated using current data

Line 42 the authors refer to the evidence of pre-symptomatic transmission, but they do not include it in their model. Why? What impact would it have?

Lines 46-48 Interventions have included a complete lockdown in several countries, so this sentence needs to be updated/changed

Lines 64-67 please add a simple explanation (few words) of how this could lead to a second wave

Methods

Line 97 Has this been shown? Reference is needed here.

Results

Line 189 The probability of infection is mentioned here for the first time. It is currently explained in Table 1 and in the Supplement, so it is worth referring to it.

Figure 3. I would remove this figure and add a similar one at the end including all scenarios investigated (for one or two choices of efficacies)

Figure 4 panel C the blue line is not visible. Explain in the caption.

Line 208 As this is not shown by the authors, it would be worth adding a ref here

Figure 5 Add caption so that the figure can stand alone

Discussion

Expected impact of pre-symptomatic transmission

[LINK]

---

## [Editor Report · Decision Letter 2]

14 May 2020

Dear Dr. Teslya,

Thank you very much for re-submitting your manuscript "Impact of self-imposed prevention measures and short-term government-imposed social distancing on mitigating and delaying a COVID-19 epidemic: A modelling study" (PMEDICINE-D-20-00973R2) for review by PLOS Medicine.

I have discussed the paper with my colleagues and the academic editor. I am pleased to say that provided the remaining editorial and production issues are dealt with we are planning to accept the paper for publication in the journal.

[LINK]

We look forward to receiving the revised manuscript by May 21 2020 11:59PM. 

Sincerely,

Adya Misra, PhD

Senior Editor 

PLOS Medicine

plosmedicine.org

Requests from Editors:

- at line 7 I'd suggest ".... compare the individual and combined effectiveness of ..."

- at line 18 I'd suggest "We estimate that a large epidemic can be prevented if ..."

- at line 20 I'd recommend "... social distancing alone is estimated to delay but not reduce the peak"

- at line 23, e.g. "Our analyses are limited in that they do not account for ..."

- that should be "adoption" at line 27

- in the author summary, I'd suggest beginning the 3rd bullet point of the "what did the authors do and find" with "We estimate that short-term government imposed ..."

- for reference 38 and other preprints I suggest the authors add "[preprint]"

- line 296 - we do not allow 'data not shown' or similar, so please remove this sentence. 

Comments from Reviewers:

[LINK]

---

## [Editor Report · Decision Letter 3]

11 Jun 2020

Dear Dr. Teslya, 

On behalf of my colleagues and the academic editor, Dr. Yuming Guo, I am delighted to inform you that your manuscript entitled "Impact of self-imposed prevention measures and short-term government-imposed social distancing on mitigating and delaying a COVID-19 epidemic: A modelling study" (PMEDICINE-D-20-00973R3) has been accepted for publication in PLOS Medicine. 

PRODUCTION PROCESS

PRESS

PROFILE INFORMATION

Thank you again for submitting the manuscript to PLOS Medicine. We look forward to publishing it. 

Best wishes, 

Adya Misra, PhD

Senior Editor 

PLOS Medicine

plosmedicine.org